# Quantized circular photogalvanic effect in Weyl semimetals

Fernando de Juan[1,2,3], Adolfo G. Grushin[1], Takahiro Morimoto[1] & Joel E. Moore[1,4]

The circular photogalvanic effect (CPGE) is the part of a photocurrent that switches depending on the sense of circular polarization of the incident light. It has been consistently observed in systems without inversion symmetry and depends on non-universal material details. Here we find that in a class of Weyl semimetals (for example, $SrSi_2$) and three-dimensional Rashba materials (for example, doped Te) without inversion and mirror symmetries, the injection contribution to the CPGE trace is effectively quantized in terms of the fundamental constants $e$, $h$, $c$ and $\epsilon_0$ with no material-dependent parameters. This is so because the CPGE directly measures the topological charge of Weyl points, and non-quantized corrections from disorder and additional bands can be small over a significant range of incident frequencies. Moreover, the magnitude of the CPGE induced by a Weyl node is relatively large, which enables the direct detection of the monopole charge with current techniques.

[1] Department of Physics, University of California, Berkeley, California 94720, USA. [2] Instituto Madrileño de Estudios Avanzados en Nanociencia (IMDEA-Nanociencia), 28049 Madrid, Spain. [3] Rudolf Peierls Centre for Theoretical Physics, Oxford OX1 3NP, UK. [4] Materials Sciences Division, Lawrence Berkeley National Laboratory, Berkeley, California 94720, USA. Correspondence and requests for materials should be addressed to F.d.J. (email: fernando.dejuan@physics.ox.ac.uk).

Wen the Fermi surface of a solid is close to a linear crossing of two bands, the low-energy quasiparticles are relativistic Weyl fermions[1–3]. This linear crossing, known as a Weyl node, is protected from becoming gapped because it carries a monopole source of Berry curvature, which leads to many unique experimental consequences. Materials with this band structure have recently been predicted[4,5] and discovered[6–9], primarily through observation in angle-resolved photoemission of an unusual surface state known as a Fermi arc. However, so far it has been challenging to find truly quantized signatures induced by the existence of the monopole, which would be analogous to the quantum Hall effect in two-dimensional systems or the half-integer Hall effect at topological insulator surfaces. The principle that quantized effects can exist in metallic systems is demonstrated by graphene, where for a broad range of frequencies the transmission of incident light is $1 - \alpha$, where $\alpha = e^2/4\pi\hbar c\epsilon_0$ is the fine structure constant[10].

A feature of Weyl fermions examined recently as a potentially quantized linear response is the anomaly induced chiral magnetic effect[11–15]: the generation of a current by an applied magnetic field. While it is now clear that there is no equilibrium current[16], a finite current is possible in the transport limit with the frequency $\omega \to 0$ after the transferred momentum $\mathbf{q} = 0$ (refs 17–19). This current has the same origin as natural optical activity[20,21]. It is determined by orbital moments rather than the chiral anomaly and its magnitude depends on a non-universal material-dependent property: the energy splitting between Weyl points. Other potential probes of the chiral anomaly are nonlinear responses to both $\mathbf{E}$ and $\mathbf{B}$ (refs 22–24), which present a characteristic angular dependence measurable in magnetotransport experiments. Current measurements show a strong angular dependence[25], but the direct relation to the chiral anomaly is subtle[26]. Other promising scattering proposals could access distinct Weyl node properties[27,28].

The main finding of this paper is that in a Weyl semimetal where nodes of opposite chirality lie at different energies, the circular photogalvanic effect (CPGE) becomes a truly quantized response that depends only on fundamental constants and the monopole charge of a Weyl node. The CPGE is the part of a DC photocurrent that switches with the sense of circular polarization. It has been measured in a variety of conventional semiconductors[29,30] and more recently in topological insulators[31,32]. The typical magnitude of the CPGE at low frequency corresponds to an observed switchable photocurrent $j \sim 10$–100 pA for incident intensity of $I \sim 1$ W over a cm-sized sample in quantum wells that have time-reversal symmetry but low spatial symmetry[29]. It has been obtained theoretically as a Berry phase effect[33–35], possibly the first in nonlinear optics, but there is no quantization: the effect measures the strength of the leading allowed Berry curvature term, which in three-dimensional (3D) materials[35] can be viewed as the dipole moment of Berry curvature.

In contrast, we find that that the CPGE induced current for a Weyl point is quantized and given by

$$\frac{1}{2}\left[\frac{\mathrm{d}j_{\circlearrowleft}}{\mathrm{d}t} - \frac{\mathrm{d}j_{\circlearrowright}}{\mathrm{d}t}\right] = \frac{2\pi e^3}{h^2 c\epsilon_0} I C_i = \frac{4\pi\alpha e}{h} I C_i, \qquad (1)$$

where $\alpha$ is the fine structure constant defined above, $C_i$ is the integer-valued topological charge of Weyl point $i$ and $I$ is the applied intensity. In this equation, the currents for left and right circular polarization $j_{\circlearrowleft}, j_{\circlearrowright}$ are perpendicular to the polarization plane, and summed over three mutually orthogonal planes. While the quantization we find is not expected to be exponentially protected as in gapped systems like in the quantum Hall effect, it is robust under small material changes in the sense that it is a direct measurement of the monopole charge in

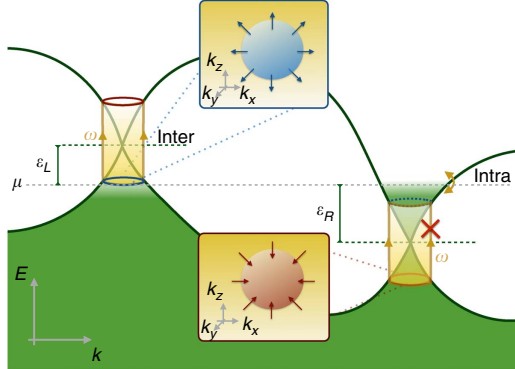

**Figure 1 | Intraband versus interband effects in Weyl semimetals.** When inversion and mirror symmetries are broken, Weyl nodes of opposite chiralities are generically at different energies. For intraband effects like optical gyrotropy, both nodes contribute and the response is proportional to the difference $\varepsilon_L - \varepsilon_R$. For an interband effect like the CPGE, when $2|\varepsilon_L| < \hbar\omega < 2|\varepsilon_R|$, one Weyl node contributes exactly with the monopole charge, while the other has zero contribution due to Pauli blocking.

units of fundamental constants, as opposed to the transparency of graphene which enjoys no such interpretation.

Equation (1) describes a current whose increase in time is proportional to intensity, known as an injection current[36]. It is generated by resonant transitions at frequency $\omega$ between the occupied valence band and the unoccupied conduction band of the Weyl node (Fig. 1). It contrasts previous finite frequency proposals[33–35,37] that originate in the low frequency response of electronic states near the Fermi level or other high-frequency[37] and interband phenomena[38] where the CPGE is not quantized. Interestingly, a CPGE was predicted for tilted Weyl nodes that lie at the same energy[39] but this effect is not quantized.

In a real material, the total Weyl node charge in the Brillouin zone must be zero[40]. Crucially, this does not preclude the observation of a finite CPGE: Weyl nodes of opposite chirality need not be at the same energy in a low-symmetry material and resonant transitions for a given node can be Pauli blocked, rendering it inactive (Fig. 1). In this case, the response is constant and quantized for a finite range of frequencies. In addition, the key fact for experimental observability is that the prefactor of equation (1) is large in comparison to ordinary CPGE magnitudes. For typical relaxation times, the quantized Weyl node contribution will dominate other metallic or insulating contributions[33–35] by more than an order of magnitude, suggesting that the total CPGE observed in experiment will indeed reveal the quantization. In what follows we analytically derive the quantized response equation (1) for two-band models and then consider corrections including those arising from additional bands. We provide supporting numerical evidence and suggest candidate materials as well as ideas for detection.

## Results

**The circular photogalvanic effect.** In materials with time reversal symmetry, an injection current can only be produced by circularly polarized light. The CPGE injection current is defined as the second order response

$$\frac{\mathrm{d}j_i}{\mathrm{d}t} = \beta_{ij}(\omega)[\mathbf{E}(\omega) \times \mathbf{E}^*(\omega)]_j, \qquad (2)$$

to an electric field $\mathbf{E}(\omega) = \mathbf{E}^*(-\omega)$, where latin indices span the cartesian components $\{x, y, z\}$. The tensor $\beta_{ij}$ is purely imaginary and only non-zero if inversion is broken and the material belongs to one of the gyrotropic point groups (The gyrotropic point

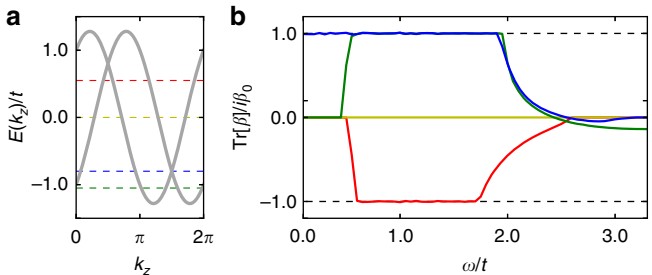

**Figure 2 | CPGE quantization for a two-band Weyl semimetal model.**
(**a**) Band structure for a generic two-band Weyl semimetal model. (**b**) CPGE trace for the same model, for four different values of the chemical potential ($\mu/t = -1.05$, $-0.8$, $0.0$, $0.55$) represented as dashed lines in **a**. For frequencies between the Weyl node energies the CPGE trace is quantized to $\beta_0 = \pi e^3/h^2$ (see main text).

groups are $C_1$, $C_2$, $C_s$, $C_{2v}$, $C_4$, $C_{4v}$, $C_3$, $C_{3v}$, $C_6$, $C_{6v}$ (ferroelectrics) and $D_2$, $D_4$, $D_{2d}$, $D_3$, $D_6$, $S_4$, $T$ and $O$ (ref. 41). Of them, the subset of enantiomorphic (or chiral) groups $C_1$, $C_2$, $C_3$, $C_4$, $C_6$, $D_2$, $D_4$, $D_3$, $D_6$, $T$ and $O$ are mirror-free and can support a quantized CPGE. The only other mirror-free group $S_4$ has an improper rotation which constrains the CPGE trace to be zero). The presence of at least one mirror symmetry constrains all the diagonal components to be zero, while the off-diagonal ones can be finite and give a non-quantized CPGE, as in ref. 39. Mirror-symmetry constrains the Weyl node at momentum **k** to have the same energy as its partner of opposite chirality at $-\mathbf{k}$, and hence the effect must vanish. The key for the quantized response to be observed is therefore that inversion and all mirror symmetries are broken, as in enantiomorphic crystals, allowing for the nodes to occur at different energies. In this case the trace of $\beta_{ij}$ is quantized for a finite range of frequencies as we proceed to show.

The CPGE tensor $\beta$ can be written in general as ref. 36

$$\beta_{ij}(\omega) = \frac{\pi e^3}{\hbar V} \epsilon_{jkl} \sum_{\mathbf{k},n,m} f_{nm}^{\mathbf{k}} \Delta_{\mathbf{k},nm}^{i} r_{\mathbf{k},nm}^{k} r_{\mathbf{k},mn}^{l} \delta(\hbar\omega - E_{\mathbf{k},mn}), \quad (3)$$

where $V$ is the sample volume, $E_{\mathbf{k},nm} = E_{\mathbf{k},n} - E_{\mathbf{k},m}$ and $f_{nm}^{\mathbf{k}} = f_n^{\mathbf{k}} - f_m^{\mathbf{k}}$ are the difference between band energies and Fermi-Dirac distributions respectively, $\mathbf{r}_{\mathbf{k},nm} = i\langle n|\partial_{\mathbf{k}}|m\rangle$ is the cross gap Berry connection and $\Delta_{\mathbf{k},nm}^{i} = \partial_{k_i} E_{\mathbf{k},nm}/\hbar$.

**Exact quantization of the CPGE for two-band models.** The position operator matrix elements in equation (3) can be related to Berry curvatures with the general expression[36]

$$\Omega_{\mathbf{k},n}^{c} = i\epsilon^{abc} \sum_{m \neq n} r_{\mathbf{k},nm}^{a} r_{\mathbf{k},mn}^{b}, \quad (4)$$

where $\Omega_n^c$ is the Berry curvature of band $n$. For a model with only two bands, this relation allows us to write

$$\beta_{ij}(\omega) = \frac{i\pi e^3}{\hbar^2 V} \sum_{\mathbf{k}} \partial_{k_i} E_{\mathbf{k},12} \Omega_{\mathbf{k}}^{j} \delta(\hbar\omega - E_{\mathbf{k},21}), \quad (5)$$

where 1, 2 correspond to valence and conduction bands, $\omega > 0$ is assumed, and $\Omega_{\mathbf{k}}^{j} \equiv \Omega_{\mathbf{k},1}^{j} = -\Omega_{\mathbf{k},2}^{j}$. At a given frequency $\omega$, the delta function selects the surface $S$ in k-space where $E_{\mathbf{k},12} = \hbar\omega$. Since by definition $\partial_{k_i} E_{\mathbf{k},12}$ is normal to this surface, the trace of $\beta$ can be written as (see Methods)

$$\mathrm{Tr}[\beta(\omega)] = i\frac{e^3}{2h^2} \oint_S d\mathbf{S} \cdot \mathbf{\Omega}, \quad (6)$$

where d$S$ denotes the oriented surface element normal to $S$. Thus the CPGE trace measures the Berry flux penetrating through $S$. In particular, when the surface $S$ surrounds a Weyl node (for example, located at $\varepsilon_L$, see Fig. 1), the above formula reduces to the monopole charge of the Weyl node, yielding a quantized CPGE

$$\mathrm{Tr}[\beta(\omega)] = i\pi\frac{e^3}{h^2} C_L \equiv i\beta_0, \quad \omega < 2\varepsilon_R, \quad (7)$$

where $C_L$ is the monopole charge of the Weyl node at $\varepsilon_L$. In terms of the applied intensity $I = \frac{c\epsilon_0}{2}|E|^2$ the quantization is given by equation (1) as anticipated. For $\omega > 2\varepsilon_R$, the second node contributes with opposite sign to $S$ and quantization is generically lost. Thus, in the ideal case of two linear Weyl nodes at energy $\varepsilon_{L,R}$ from the chemical potential $\mu$ the quantization holds as long as $2|\varepsilon_R| > \omega > 2|\varepsilon_L|$ and $\varepsilon_L \neq \varepsilon_R$. For isotropic Weyl fermions (that is, linear dispersion with isotropic Fermi velocity and no tilting), measuring only one component of CPGE already suffices since $\beta_{xx} = \beta_{yy} = \beta_{zz} = i\beta_0/3$.

To support these findings we have numerically calculated the injection current for a two-band model with a characteristic energy scale $t$ (see Methods). Our results are summarized in Fig. 2a) shows the band structure for representative parameters as a function of the momentum along separating the Weyl nodes ($k_z$). The dashed lines outline four different chemical potentials $\mu$ for which the injection current is calculated using equation (3) and shown in panel b). Consistent with our discussion, when the chemical potential is chosen such that $\varepsilon_L = -\varepsilon_R$ the CPGE is zero (orange flat-line). When $\mu$ coincides with the right-most node (blue dashed line) $\varepsilon_R = 0$ and the CPGE is quantized to $\beta_0$ from $\omega = 0$. Note that although in the idealized Weyl semimetal model quantization is expected to hold up to $\omega = 2\varepsilon_L$, in a lattice model $2\varepsilon_L$ can exceed the band width. This is the case of all non-trivial cases in Fig. 2 and thus the quantization disappears at a frequency $\omega \lesssim 2\varepsilon_L$. With this caveat, for all generic choices of parameters the CPGE is numerically quantized consistent with our analytics.

**Higher band corrections.** In practice, corrections from higher bands can lead to a non-universal CPGE since the CPGE can only be written exactly as a Berry curvature flux for two-band models.

To quantify the importance of these corrections consider a three band model with two lower bands forming the Weyl nodes as above that are complemented by a third unoccupied band. Using that $r_{\mathbf{k},nm}^{a} = -iv_{\mathbf{k},nm}^{a}/E_{\mathbf{k},nm}$ it is possible to rewrite equation (3) as $\beta(\omega) = i\beta_0 + \delta\beta(\omega)$ for small $\omega$ (see Methods). These corrections become arbitrarily small when $\omega \to 0$ for $\mu = 0$ because $\mathbf{v}_{\mathbf{k},nm} \sim (\partial_{\mathbf{k}}H)_{mn}$ remains a non-singular function for any pair of bands, while $E_{\mathbf{k},nm}$ is arbitrarily small for the two bands at resonance forming the Weyl node. Explicitly, the correction scales as

$$\delta\beta(\omega) \propto \frac{|i\mathbf{v}_{\mathbf{k},13} \times \mathbf{v}_{\mathbf{k},13}|}{v_F^2} \frac{\omega^2}{E_{13}^2}, \quad \omega/t \ll 1, \quad (8)$$

where $v_F$ is a characteristic Fermi velocity around a Weyl node and $E_{13}$ is the typical energy difference between the occupied first band and the unoccupied third band. The corrections vanish as $\omega^2$ and are inversely proportional to the energy separation to higher bands, thus becoming unimportant at low enough frequencies. Note as well that the matrix elements in equation (8) are typically small for different orbitals rendering the departure from quantization even less observable in practice.

We have assessed numerically the effect of higher bands on quantization by calculating the CPGE of a generic four-band model[16]. This model can describe Weyl semimetals with nodes at different energies such as SrSi$_2$ (ref. 42; see Fig. 3, top left)

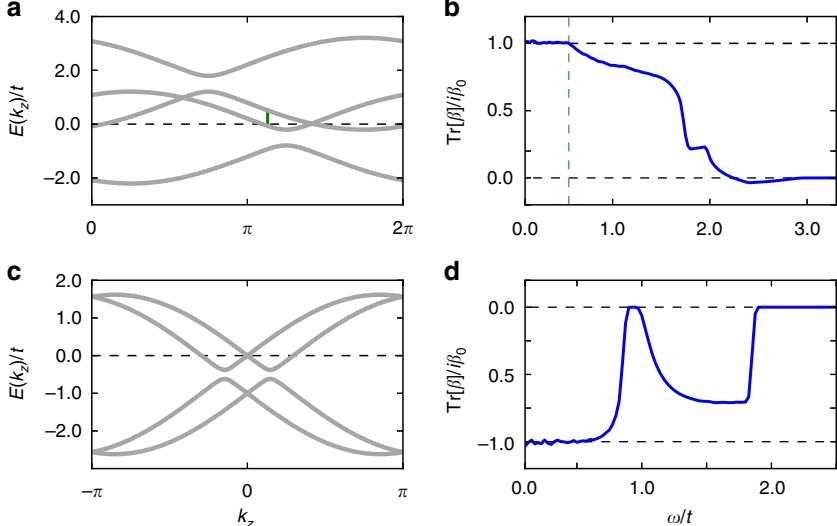

**Figure 3 | CPGE for four-band models.** (**a**) Band structure for a four-band Weyl semimetal with broken inversion symmetry. (**b**) CPGE trace for same model, for chemical potential shown as a dashed line in **a**. (**c,d**) The same for a model of a 3D Rashba material. Both models show a quantized injection current for small frequencies. The dashed vertical line in **b** corresponds to the frequency $\omega/t \sim 0.6$ above which additional transitions, denoted by a solid vertical line in **a**, that preclude quantization are allowed.

relevant for our purposes and, with straightforward modifications, Dirac semimetals. It can also describe materials where the band edge takes the form of a 3D Rashba-like Hamiltonian[43] $H_{\mathbf{k}} = \mathbf{k}^2/2m + \lambda\boldsymbol{\sigma}\cdot\mathbf{k}$. This is the natural spin-orbit splitting of parabolic bands in the absence of inversion and mirror symmetries. It generates a single Weyl node near the band edge, and the node of opposite chirality naturally appears at significantly different energies (Fig. 3c). Since the contribution of the outer Fermi surface to the CPGE is expected to be zero, a 3D Rashba material can show a quantized CPGE, and concrete examples are discussed below.

In Fig. 3 we show the injection current for two representative band structures: a Weyl semimetal with broken inversion symmetry and a 3D Rashba material. In all plots $\mu$ coincides with a Weyl node; other choices behave qualitatively as described in Fig. 2. Despite the presence of higher bands, the quantization is robust for small frequencies for all studied cases. We find this encouraging for experiments from both our numerical and analytic analysis that predict that the corrections due to higher bands are expected to be small.

## Discussion

We now elaborate on the main practical aspects that suggest that the quantized CPGE can be observed with current experiments. In the absence of any scattering mechanism, equation (2) predicts a quantized rate of unbounded current growth. In practice, disorder will introduce a finite scattering rate $1/\tau$ and the linear growth of current can only be observed for times $t < \tau$, which in existing Weyl semimetals is $\tau \sim 1\,\mathrm{ps}$ (refs 23,44). In the limit of $t \gg \tau$, the current saturates to $j^{\mathrm{sat}}$ that can be computed from Fermi's golden rule or with Floquet theory (see Methods), resulting in

$$j_i^{\mathrm{sat}} = \tau\beta_{ij}(\omega)(\mathbf{E}(\omega)\times\mathbf{E}^*(\omega))_j, \qquad (9)$$

with $\beta_{ij}(\omega)$ defined as above. The total stationary photocurrent will therefore be $\frac{1}{2}[j_{\circlearrowleft}^{\mathrm{sat}} - j_{\circlearrowright}^{\mathrm{sat}}] = \frac{2\pi e^3}{h^2 c\varepsilon_0}I\tau$, with the universal coefficient $\frac{2\pi e^3}{h^2 c\varepsilon_0} = 22.2\frac{A}{W\,ps}$. Note this assumes the intensity $I$ should remain constant throughout the sample, so light absorption should be small. The attenuation depth of a Weyl semimetal scales as

$\delta \sim 1/\omega$, and we estimate that for $\omega < 25\,\mathrm{THz}$, $\delta > 1\,\mathrm{\mu m}$ (see Methods). Absorption is thus negligible for typical thin film dimensions. Taking a thickness of 10 nm, an area of $1\,\mathrm{mm}^2$, and an irradiation time of 1 ps ($\sim\tau$), the induced photocurrent reaches $j^{\mathrm{sat}} \simeq 2\,\mathrm{nA}/(W/\mathrm{cm}^2)$. This is much larger than the reported CPGE current of 10–100 pA for a topological insulator thin film[32], and thus its measurement is experimentally feasible. The additional Fermi surface contribution that can be described semiclassically[33] is estimated to be much smaller $\left[\simeq 10\,\mathrm{pA}/(W/\mathrm{cm}^2)\right]$ so that the quantized CPGE contribution is dominant. Surface corrections can also be neglected at normal incidence, since the expected current is normal to the surface, and are in any case small compared to the bulk CPGE we describe.

Experimentally, the rate of current injection can be extracted from an all-optical setup with no free parameters as in ref. 45. There, direct time-resolved measurements of photocurrent are possible using short pulses of intense light. The time-dependent photocurrent can be measured as a radiated signal of low frequency set by the envelope of the incident pulses. Alternatively, if only the simpler steady-state measurement is available, the relaxation time $\tau$ can be estimated from other measurements such as the broadening of the drop at $2\mu$ in the linear optical conductivity or from the CPGE itself by measuring the width of the jump at $\hbar\omega \sim \min(\varepsilon_L, \varepsilon_R)$. The measured value of $\tau$ could be divided into the photocurrent to get the universal CPGE quantum.

Observing quantization requires a Weyl semimetal where inversion and all mirror symmetries are absent. The recently realized inversion breaking Weyl semimetals in the monopnictide class, such as TaAs (refs 6,7), do have a mirror plane in their structure. Shear strain for example can break this symmetry, opening a small window of frequencies to observe the effect. A better candidate is $SrSi_2$ (ref. 42): all mirror symmetries are broken, the Weyl nodes of opposite chiralities are separated significantly in energy ($\sim 0.1\,\mathrm{eV}$) and the chemical potential is close to one of the Weyl nodes. Other material candidates for mirror-free Weyl semimetals have been recently predicted in ref. 46. As we have shown, the quantized CPGE can also be observed in 3D Rashba materials[43] as in the conduction band of trigonal elemental Te (ref. 47). Note that BiTeI (ref. 48) does not

have a 3D Rashba band structure despite its strong spin-orbit splitting because of its mirror symmetries, which could also be broken by strain. Synthetic 3D Rashba materials can be also engineered in cold atoms[43] which can be driven periodically to study the effects presented here.

The quantization of the CPGE is not limited to linear nodes. It can occur for any node with $C>1$ as long as the node is formed only by two bands[49,50] and equation (4) applies, as it happens in SrSi$_2$ where $C=2$. Nodal crossings with three or more bands[51], however, are not expected to display quantization of this type as the corrections from equation (8) cannot be made small. We also note that the quantized value of the CPGE response is independent of any tilting of the nodes as long as they remain of type I, but the frequency window to observe it will depend on the tilt parameter[52]. If the tilting is strong enough to create a type II node[53,54], the surface of allowed transitions encompasses only a fraction of the sphere surrounding the node and the quantization is lost at all frequencies. We also note that, unlike optical gyrotropy which is non-quantized and allowed for any metal with broken inversion, the quantized CPGE can occur only in the presence of Weyl nodes.

In conclusion, we have shown that the trace of the circular photogalvanic tensor is quantized for Weyl semimetals and 3D Rashba materials that break inversion and all mirror symmetries. We have identified several candidate materials to observe this effect, which we estimate to be an order of magnitude larger compared to other more conventional contributions.

## Methods

**Analytical computation of CPGE coefficient.** The CPGE tensor $\beta_{ij}$ for a two band model is given by equation (5) in the main text. The trace of this tensor is

$$\text{Tr}[\beta(\omega)] = \frac{i\pi e^3}{\hbar^2 V} \sum_{\mathbf{k}} \partial_{k_i} E_{\mathbf{k},12} \Omega^i_{\mathbf{k}} \delta(\hbar\omega - E_{\mathbf{k},12}). \quad (10)$$

To perform the integral we use that for an isotropic Weyl node $\partial_{k_i} E_{\mathbf{k},12} = 2v_F k^i/k$ where $k=|\mathbf{k}|$ and therefore $\delta(\hbar\omega - E_{\mathbf{k},12}) = \delta[k-k(\omega)]/(2v_F)$, where $k(\omega) = \omega/2v_F$, and that Berry curvature of such Weyl node is given by $\Omega^i = \frac{1}{2}k^i/k^3$. We then get

$$\text{Tr}[\beta(\omega)] = \frac{e^3\pi}{\hbar^2} i \int \frac{d\Omega}{(2\pi)^3} \int k^2 dk \frac{2v_F k_i}{k} \frac{1}{2} \frac{k_i}{k^3} \frac{\delta(k-k(\omega))}{2v_F}$$
$$= i\frac{e^3\pi}{\hbar^2} \frac{4\pi}{2(2\pi)^3} = i\frac{e^3\pi}{\hbar^2}. \quad (11)$$

To relate this response coefficient to the applied intensity, we consider circularly polarized light for which $[\mathbf{E}(\omega) \times \mathbf{E}^*(\omega)]_j = i|\mathbf{E}|^2 n_j$ with $n_j$ a unit vector normal to the polarization plane. For the $x-y$ plane, for example, we have $\mathbf{E} = |E|(1, i, 0)/\sqrt{2}$ and $n_j = (0, 0, 1)$. From equation (2) the injection current induced in the $z$ direction is given by

$$\partial_t j_z = \beta_{zz} i|\mathbf{E}|^2. \quad (12)$$

To get the trace, we add up the contributions from the three orthogonal directions, defining $\partial_t j_\circlearrowleft = \left(\beta_{xx} + \beta_{yy} + \beta_{zz}\right) i|\mathbf{E}|^2$, and use $I = c\varepsilon_0|\mathbf{E}|^2/2$

$$\partial_t j_\circlearrowleft = \frac{e^3\pi}{h^2}|\mathbf{E}|^2 = \frac{e^3 2\pi}{h^2 c\varepsilon_0} I = 4\pi\alpha \frac{e}{h} I, \quad (13)$$

in terms of the fine structure constant $\alpha = e^2/(4\pi\epsilon_0\hbar c)$. The saturation current density with finite lifetime $\tau$ is simply

$$j_\circlearrowleft^{\text{sat}} = 4\pi\alpha \frac{e}{h} \tau I = 22.17 \frac{\tau}{\text{ps}} \frac{I}{W/\text{cm}^2} \frac{A}{\text{cm}^2}. \quad (14)$$

Finally, note this quantity is by construction the one that reverses sign when circular polarization is reversed. In practice other contributions that do not change sign exist in addition to the quantized CPGE. These can be removed by taking $(\partial_t j_\circlearrowleft - \partial_t j_\circlearrowright)/2$ or $\left(j_\circlearrowleft^{\text{sat}} - j_\circlearrowright^{\text{sat}}\right)/2$ as in the main text.

**Absorption and attenuation length.** When light is irradiated in any conducting material, the intensity decays exponentially from the surface $I = I_0 e^{-\alpha x}$ due to light absorption. The attenuation constant is expressed in terms of the dielectric function as

$$\alpha = \omega/c \sqrt{2(-\text{Re}[\epsilon] + |\epsilon|)} \quad (15)$$

which is related to conductivity by $\epsilon = 1 - \frac{4\pi\sigma(\omega)}{i\omega}$. The conductivity of a Weyl semimetal for $\omega \gg \mu$ is given by $\text{Re}[\sigma] = \frac{e^2}{24\pi v_F\hbar}\omega$ (ref. 55), which gives an

attenuation length $\delta = 1/\alpha$

$$\delta = \lambda \frac{1}{2\pi\sqrt{2\left(-1 + \sqrt{1 + \left(\alpha \frac{c}{6v_F}\right)^2}\right)}} \sim 0.23\lambda \quad (16)$$

where we have used a typical Fermi velocity $v_F = 5 \times 10^5 \text{ m s}^{-1}$ (ref. 44). For frequencies below $\omega = 100 \text{ meV}$ ($\nu = 25 \text{ THz}$), we have $\lambda = 12 \text{ μm}$ and $1/\alpha = 2.7 \text{ μm}$ so absorption is negligible for thin films in the THz range.

**Higher band corrections.** In this section we discuss how the quantization is modified by the presence of higher bands. We consider the case of three bands: Bands 1 and 2 host the Weyl nodes while we choose band 3 to be higher in energy and unoccupied. The CPGE coefficient is explicitly

$$\text{Tr}[\beta] = \frac{\pi e^3}{\hbar V} \sum_{\mathbf{k}} \epsilon^{abc} \Big[ \Delta^a_{\mathbf{k},12} r^b_{\mathbf{k},12} r^c_{\mathbf{k},21} \delta(\hbar\omega - E_{\mathbf{k},12})$$
$$+ \Delta^a_{\mathbf{k},13} r^b_{\mathbf{k},13} r^c_{\mathbf{k},31} \delta(\hbar\omega - E_{\mathbf{k},13}) \Big]. \quad (17)$$

If we assume that the probing frequencies are always smaller than $E_{\mathbf{k},13}$ then $\delta(\hbar\omega - E_{\mathbf{k},13})$ will never contribute and can be discarded. For this case

$$\text{Tr}[\beta] = \frac{\pi e^3}{\hbar V} \sum_{\mathbf{k}} \epsilon^{abc} \Delta^a_{\mathbf{k},12} r^b_{\mathbf{k},12} r^c_{\mathbf{k},21} \delta(\hbar\omega - E_{\mathbf{k},12}). \quad (18)$$

The existence of an extra band now modifies the sum rules to

$$\Omega^c_{\mathbf{k},1} = i\epsilon^{abc}\left(r^a_{\mathbf{k},12} r^b_{21} + r^a_{\mathbf{k},13} r^b_{31}\right), \quad (19)$$

$$\Omega^c_{\mathbf{k},2} = i\epsilon^{abc}\left(r^a_{\mathbf{k},21} r^b_{\mathbf{k},12} + r^a_{\mathbf{k},23} r^b_{\mathbf{k},32}\right), \quad (20)$$

we may write

$$\text{Tr}[\beta] = -\frac{\pi e^3}{\hbar V} \sum_{\mathbf{k}} \Delta^a_{\mathbf{k},12}\left(i\Omega^a_1 + \epsilon^{abc} r^b_{\mathbf{k},13} r^c_{\mathbf{k},31}\right) \delta(\hbar\omega - E_{\mathbf{k},12}). \quad (21)$$

Using that $r^a_{\mathbf{k},nm} = -iv^a_{\mathbf{k},nm}/E_{\mathbf{k},nm}$ with $v^a_{\mathbf{k},nm} = (\partial_a H)_{\mathbf{k},nm}$ the quantization will be preserved if $i\Omega^a_{\mathbf{k},1} \gg \epsilon^{abc} v^b_{\mathbf{k},13} v^c_{\mathbf{k},31}/E^2_{\mathbf{k},13}$ for every direction $a$. To make a quantitative estimate we take an isotropic Weyl node with $E_{\mathbf{k}} = v_F|\mathbf{k}|$ and $\Omega_{\mathbf{k}} = \mathbf{k}/|\mathbf{k}|^3$. The correction to the quantized value can be estimated as the dimensionless ratio of the moduli of the two vectors inside the parenthesis

$$\delta\beta \propto \frac{|i\mathbf{v}_{\mathbf{k},13} \times \mathbf{v}_{\mathbf{k},13}|}{E^2_{\mathbf{k},13}/|\mathbf{k}|^2}. \quad (22)$$

Assuming zero chemical potential and small probing frequency and using that around the Weyl node we have $k^2 = \omega^2/v_F^2$ we obtain

$$\delta\beta \propto \frac{|i\mathbf{v}_{\mathbf{k},13} \times \mathbf{v}_{\mathbf{k},13}|}{v_F^2} \frac{\omega^2}{E^2_{13}}. \quad (23)$$

Therefore at low frequencies, the corrections to the quantization of $\text{Tr}[\beta]$ vanish quadratically, since $v^a_{nm}$ is a derivative of the Hamiltonian and cannot be singular at the node.

**Lattice models.** In the main text we have used a two and a four band lattice model which we describe here with more detail. The two band model is defined by $\mathcal{H}_{\mathbf{k}} = \mathbf{d}_{\mathbf{k}} \cdot \boldsymbol{\sigma} + \varepsilon_{\mathbf{k}}\sigma^0$ with

$$\mathbf{d}_{\mathbf{k}} = -\left\{ t\sin k_x, t\sin k_y, -M + t\sum_{i=x,y,z}\cos k_i \right\}, \quad (24)$$

$$\varepsilon_{\mathbf{k}} = \gamma\sin(k_z), \quad (25)$$

where $\sigma^0$ is the $2 \times 2$ identity matrix and $\boldsymbol{\sigma} = (\sigma^x, \sigma^y, \sigma^z)$ the Pauli matrices. For $1 < |M/t| < 3$ it has a pair of Weyl cones at $\mathbf{k} = \{0, 0, \pm K_0\}$ with $K_0 = \cos^{-1}(M/t - 2)$ at energies $\epsilon_\pm = \pm\gamma\sin(K_0)$. The band structure shown in Fig. 2 corresponds to for $M/J = 2$ and $\gamma/t = 0.8$. The chemical potential can be controlled by adding a constant term proportional to $\mu\sigma_0$. Note that both inversion and time reversal symmetry are broken in this model. By doubling the model one can restore time reversal symmetry while still being inversion odd. Since multiple copies of the model defined by equations (24) and (25) will only result in an additional prefactor in equation (1) proportional to the number of optically active Weyl nodes, in the main text we use the model defined by equations (24) and (25).

The second model that we use to investigate the effect of higher bands is a four band model that can originate from an orbital degree of freedom $A$, $B$ and spin $\uparrow$, $\downarrow$. The Hamiltonian $H_{4b}$ in the basis defined by the electron operator $c_{\mathbf{r}} = \left(c_{\mathbf{r}A\uparrow}, c_{\mathbf{r}A\downarrow}, c_{\mathbf{r}B\uparrow}, c_{\mathbf{r}B\downarrow}\right)^T$ is the sum of three terms[16]

$$H_{4b} = \sum_{\mathbf{k},j} D^j(\mathbf{k}) c_{\mathbf{k}}^\dagger \Gamma^j c_{\mathbf{k}} + b^j c_{\mathbf{k}}^\dagger \Gamma_b \Gamma^j c_{\mathbf{k}} + b_0 c_{\mathbf{k}}^\dagger \Gamma_b c_{\mathbf{k}}. \quad (26)$$

For a detailed discussion of the phase diagram of this model we refer the reader to refs 16,56. Here we will highlight the aspects that are relevant to the calculation in the main text.

The first term describes in general a trivial or topological (either weak or strong) topological insulator and respects both time-reversal symmetry ($\mathcal{T}$) and inversion symmetry ($\mathcal{I}$). It is defined through the $\Gamma$-matrices $\Gamma^j = (\sigma^z s^y, \sigma^z s^x, \sigma^y s^0, \sigma^x s^0)$ and

$$D^j(\mathbf{k}) = -\left( t\sin k_x, t\sin k_y, t\sin k_z, t\sum_i \cos k_i - M \right), \quad (27)$$

in the subspace where the Pauli matrices $\boldsymbol{\sigma} = (\sigma^x, \sigma^y, \sigma^z)$ and $\mathbf{s} = (s^x, s^y, s^z)$ act on orbital and spin degrees of freedom respectively and $\sigma^0$ and $s^0$ are identity matrices in the corresponding subspace. The transition between trivial or topological insulator phases is governed by $M/t$. In the main text we have set $M/t = 2.5$, such that at $\mathbf{k} = 0$ the insulating state corresponds to a strong topological insulator.

The second term defined via the matrix $\Gamma_b = \sigma^y s^z$ and constant vector $\mathbf{b} = (b^x, b^y, b^z)$ breaks $\mathcal{T}$ and drives the transition to a Weyl semimetal. The absolute value of b controls the distance between Weyl nodes. In the main text we have set $\mathbf{b} = (0, 0, t)$ for Fig. 3 top row and $\mathbf{b} = (0, 0, 0)$ in Fig. 3 bottom row.

The last term is defined via the constant scalar $b_0$. It breaks $\mathcal{I}$ and separates the two Weyl nodes in energy. In all of Fig. 3 we set $b_0/t = 0.5$. As for the two-band model the chemical potential is controlled by adding a constant term proportional to $\mu\sigma_0 s_0$ and time reversal symmetry can be restored by an appropriate doubling of the model.

**Floquet theory derivation for CPGE.** In this section, we present an alternative derivation of the stationary photocurrent proportional to the relaxation time $\tau$ by using the Floquet theory. We again consider the two band model for the Weyl fermion defined by $\mathcal{H}_{\mathbf{k}}$, where energy dispersions of the valence and conduction bands are given by $E_{\mathbf{k},1}$ and $E_{\mathbf{k},2}$, respectively. In order to study dc current induced by photoexcitation between the two bands, we study the Floquet two band model consisting of the one photon dressed valence band and the bare conduction band, which is given by ref. 57

$$H_F = \begin{pmatrix} E_{\mathbf{k},1} + \hbar\omega & -ieA^*\left[ v^x_{\mathbf{k},12} - iv^y_{\mathbf{k},12} \right] \\ ieA\left[ v^x_{\mathbf{k},21} + iv^y_{\mathbf{k},21} \right] & E_{\mathbf{k},2} \end{pmatrix} \quad (28)$$
$$\equiv d_0 + \mathbf{d}\cdot\boldsymbol{\sigma},$$

with $A = E/\omega$ and the velocity matrix element $v^i_{\mathbf{k},nm} = (1/\hbar)\langle n|\partial_{k_i} H_{\mathbf{k}}|m\rangle$. These Floquet bands show anticrossing at the optical resonance at $E_{\mathbf{k},2} - E_{\mathbf{k},1} = \hbar\omega$ which describe steady state under driving. The occupation of the Floquet bands is determined by coupling to a heat bath which we assume to have the Fermi energy between the valence and conduction band. This enables us to compute steady dc current by using Keldysh Green's function method. Namely, by using the current operator along the z direction in the Floquet formalism,

$$\tilde{v}^z = \frac{1}{\hbar}\frac{\partial H_F}{\partial k_z} \equiv b_0 + \mathbf{b}\cdot\boldsymbol{\sigma}, \quad (29)$$

the dc current in the steady state is given by ref. 57

$$J = e\int \frac{d\mathbf{k}}{(2\pi)^3}(j_1 + j_2 + j_3), \quad (30)$$

with

$$j_1 = \frac{\frac{\Gamma}{2}(-d_x b_y + d_y b_x)}{d^2 + \frac{\Gamma^2}{4}}, \quad (31)$$

$$j_2 = \frac{(d_x b_x + d_y b_y)d_z}{d^2 + \frac{\Gamma^2}{4}}, \quad (32)$$

$$j_3 = \frac{\left(d_z^2 + \frac{\Gamma^2}{4}\right)b_z}{d^2 + \frac{\Gamma^2}{4}} + b_0. \quad (33)$$

The $j_1$ term describes the shift current in the case of linearly polarized light. The $j_2$ term does not lead to the current response proportional to relaxation time; while the factor $d_z/(d^2 + \frac{\Gamma^2}{4})$ result in the factor $\tau(k-k_0)\delta(k-k_0)$ with the resonant wave number $k_0$, this contribution vanishes after $k$-integration. The $j_3$ term gives the injection current if we consider the term proportional to $|E|^2$. In the following, we focus on the $j_3$ term and derive the Berry curvature formula for the injection current. The injection current $J_{inj}$ is obtained by expanding the $j_3$ term up to $A^2$ as

$$J_{inj} = -e^3|A|^2 \int \frac{d\mathbf{k}}{(2\pi)^3} \frac{\left| v^x_{\mathbf{k},12} - iv^y_{\mathbf{k},12} \right|^2}{d_z^2 + \frac{\Gamma^2}{4}} b_z. \quad (34)$$

We note that the $O(|A|^0)$ term in $j_3$ vanishes after $k$-integral due to the band connectivity. By noticing

$$\left| v^x_{\mathbf{k},12} - iv^y_{\mathbf{k},12} \right|^2 = \left| v^x_{\mathbf{k},12} \right|^2 + \left| v^y_{\mathbf{k},12} \right|^2 - 2\text{Im}\left[ v^x_{\mathbf{k},12} v^y_{\mathbf{k},21} \right], \quad (35)$$

we can write

$$J_{inj} = -\frac{2\pi\tau e^3|A|^2}{\hbar}\int \frac{d\mathbf{k}}{(2\pi)^3}\left[ v^z_{\mathbf{k},11} - v^z_{\mathbf{k},22} \right]\delta(d_z) \\ \times \left[ \left| v^x_{\mathbf{k},12} \right|^2 + \left| v^y_{\mathbf{k},12} \right|^2 - 2\text{Im}\left[ v^x_{\mathbf{k},12} v^y_{\mathbf{k},21} \right] \right], \quad (36)$$

where $\tau = \hbar/\Gamma$ is the relaxation time and $\delta(x) = \lim_{a\to 0}(1/\pi)a/(x^2 + a^2)$. Since the integrand of the first term is odd under the time-reversal symmetry, the first term vanishes after $k$-integration. The second term is described by the Berry curvature $\Omega^z_{\mathbf{k}}$ by using the identity

$$\Omega^z_{\mathbf{k}} = -2\frac{\text{Im}\left[ v^x_{\mathbf{k},12} v^y_{\mathbf{k},21} \right]}{(E_{\mathbf{k},1} - E_{\mathbf{k},2})^2}. \quad (37)$$

Thus we obtain the injection current in time reversal-symmetric systems as

$$J_{inj} = -\frac{2\pi e^3\tau|E|^2}{\hbar}\int \frac{d\mathbf{k}}{(2\pi)^3}\left[ v^z_{\mathbf{k},11} - v^z_{\mathbf{k},22} \right]\Omega^z_{\mathbf{k}}\delta(E_{\mathbf{k},12} + \hbar\omega). \quad (38)$$

Alternatively, since the Berry curvatures for the valence and conduction bands satisfy the relation $\Omega^z = \Omega^z_1 = -\Omega^z_2$, this can be rewritten as

$$J_{inj} = -\frac{e^3\tau|E|^2}{h^2}\int d\mathbf{k}\left[ \frac{\partial(E_{\mathbf{k},1} - E_{\mathbf{k},2})}{\partial k_z}\Omega^z_{\mathbf{k}} \right]\delta(E_{\mathbf{k},12} + \hbar\omega). \quad (39)$$

Namely, the nonlinear coefficient $\beta$ (in $J_i = \tau\beta_{ij}[\mathbf{E}\times\mathbf{E}^*]_j$) is given by

$$\beta_{zz} = i\frac{e^3}{2h^2}\int d\mathbf{k}\left[ \frac{\partial(E_{\mathbf{k},1} - E_{\mathbf{k},2})}{\partial k_z}\Omega^z_{\mathbf{k}} \right]\delta(E_{\mathbf{k},12} + \hbar\omega). \quad (40)$$

Thus the sum of the nonlinear conductivities $\text{Tr}[\beta] = \beta_{xx} + \beta_{yy} + \beta_{zz}$ is described by the Berry flux over the surface $S$ of the resonance condition in $k$-space as

$$\text{Tr}[\beta] = i\frac{e^3}{2h^2}\int_S d\mathbf{S}\cdot\boldsymbol{\Omega}_{\mathbf{k}}, \quad (41)$$

where $d\mathbf{S}$ denotes the oriented surface element normal to $S$. When the surface $S$ surrounds a Weyl point, this leads to quantized injection current as

$$\text{Tr}[\beta] = i\frac{\pi e^3}{h^2}. \quad (42)$$

**Data availability.** The data that support the findings of this study are available from the corresponding author upon request.

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

## Acknowledgements

We acknowledge useful discussions with F. Flicker, B. Fregoso, K. Landsteiner, N. Nagaosa, T. Neupert, J. Orenstein, G. Refael, I. Souza and M.A.H. Vozmediano. This work was supported by the Marie Curie Programme under EC Grant agreements No. 653846 (A.G.G.) and No. 705968 (F.d.J.), by the European Research Council through Grant No. 290846 (F.d.J.), by the Gordon and Betty Moore Foundation's EPiQS Initiative Theory Center Grant (T.M.), and NSF DMR-1507141 and a Simons Investigatorship (J.E.M.).

## Author contributions

All authors contributed equally to the results presented in this work and the writing of the manuscript.

## Additional information

**Competing interests:** The authors declare no competing financial interests.

**Publisher's note**: 

