## [Peer Review File · Nature Communications]

Reviewers' comments:

Reviewer #1 (Remarks to the Author):

The manuscript by Fernando de Juan et al report a prediction of a quantized response in the circular photogalvanic effect (CPGE). Specifically, the authors predict that, in a Weyl semimetal with no mirror symmetry and no inversion symmetry, the CPGE trace is quantized. Prior to this work, no one has considered a quantized CPGE response. Furthermore, the quantized CPGE can be used to measure the monopole charge of a Weyl node. Therefore, the theoretical proposal appears to be novel and important. On the other hand, the following issues need to be addressed by the authors.

1. Scope of application: The authors should clarify the scope within which their theory applies. More specifically, the questions are listed below

1.1. Does the theory only works for single Weyl nodes (± 1 chiral charge) ?

1.2. Or, does it also work for double Weyl nodes (± 2 chiral charge)? The authors mentioned SrSi₂, which has double Weyl nodes, but they didn't justify whether the higher chiral charge affects their theory or not.

1.3. Furthermore, does the theory also work for crossings with a higher degeneracy (3-band crossings, 4-band crossings, 6-band crossings, etc)? Recently, these higher degeneracy band crossings have been proposed (e.g., Bradlyn et al., Science DOI: 10.1126/science.aaf5037).

2. Experimental feasibility: As indicated from equation 1 ($\frac{1}{2} [(dJ_{RCP})/dt - (dJ_{LCP})/dt] = 4n\alpha e/h IC$), the quantization still depends on the intensity of the laser, I. This is quite difficult to measure in experiment. It depends on multiple factors, the absorption coefficient, the thickness of the sample, the size of the beam spot, etc. Because the purpose is to measure a quantized value, it becomes important to determine I precisely. The authors should comment on this aspect.

3. Lastly, a minor issue. The proposal requires a Weyl semimetal with no mirror and no inversion. This type of Weyl semimetal has been proposed and systematically analyzed in Chang et al., <https://arxiv.org/abs/1611.07925> with more material candidates. The authors should consider citing this paper.

Reviewer #2 (Remarks to the Author):

In the present manuscript the authors find a quantized value for the injection current (photocurrent) part of the circular photogalvanic effect in Weyl semimetals (WSM hereafter) when the Weyl nodes are at different energies in the Brillouin Zone: for a (non-universal and material dependent) set of frequencies, the photocurrent part saturates to the value dictated by the topological charge associated to the Weyl nodes.

As a whole, I think that the content of the present paper can be relevant for the field of Weyl semimetals: as the authors mention, so far, there is no other way to measure by optical means the topological charge associated to the Weyl nodes in time reversal invariant WSM (ideally, in time reversal breaking WSM, the quantum anomalous Hall current contains this precise information). The claim offered by this manuscript might attract the interest of the broad audience interested in WSM, and to other researchers not primarily interested in WSM but interested in non-linear optical responses. In view of this, the manuscript deserves to be published in a platform that gives to it enough visibility, as Nature Communications, if the authors address adequately the following questions I have about the content of the manuscript:

1. The authors claim in the title and in the abstract that they have found a quantized circular

photogalvanic effect in WSM. What they have actually found is a quantized value of a piece of the circular photogalvanic effect (CPGE): the photocurrent or injection current. There are other components that are part of the CPGE and they are taking a quantized value. While this statement is made clear throughout the body of the manuscript, it is not explicitly written in the title and abstract. It is mentioned that the trace part of the effect is quantized, but, for the benefit of the readers not being experts in these non-linear effect, it should be explicitly said in the title or abstract.

2. The authors compute the injection current within the dipole approximation, so there is momentum conservation together with energy conservation. It means that, when computing the coefficient β_{ii} with the Fermi golden's rule (or Floquet), the imaginary part comes from a resonant (or nesting) condition between the valence and conduction surfaces, and this precise sum over momenta is what allows the authors to find the integral over the Berry curvature (over the whole Fermi surface). They also mention a proposed material, SrSi₂. But looking at the bandstructure, it seems that the Weyl nodes are tilted, even very close to the Weyl node. This motivates this question: when tilting is unavoidable, in Weyl semimetals the optical conductivity at energies of two times the chemical potential does not take a quantized value (for isotropic WSM it does, leading to a quantized value of the conductivity at this energy). The reason for this is that, due to the tilting, the nesting condition is lost and the sum over momenta does not sweep the whole Fermi surface but just one dimensional lines, at best. This has been addressed recently (PHYSICAL REVIEW B 94, 165111 (2016)). My question then is, when tilting is unavoidable, if the same effect happens here, and the sum over the whole Fermi surface leading to the value of the topological charge disappears, leading to a non quantized value of β .

3. Even for inversion symmetric systems, the presence of surfaces in real samples trivially breaks inversion symmetry, inducing second-order non linear optical responses. The question is if the presence of surfaces can contribute to the injection current leading to a non-quantized value of β . Can the effect of the surface be ruled out in the injection current part?

4. A (really) minor stylist suggestion: When computing the injection current by using Floquet, the authors use the same notation for the velocity elements and the position of the Weyl nodes in the previous section. I suggest the authors to clarify the nomenclature.

5. The lattice models the authors employ have been used and described much before the reference [49], for instance, in ref.[16] and many others. I would encourage the authors to be "fair" and make an effort to give credit to other publications discussing these models.

Response to Referees:

Referee A

- (1.1. Does the theory only works for single Weyl nodes (± 1 chiral charge) ? 1.2. Or, does it also work for double Weyl nodes (± 2 chiral charge)? The authors mentioned SrSi₂, which has double Weyl nodes, but they didn't justify whether the higher chiral charge affects their theory or not. 1.3. Furthermore, does the theory also work for crossings with a higher degeneracy (3-band crossings, 4-band crossings, 6-band crossings, etc)? Recently, these higher degeneracy band crossings have been proposed (e.g., Bradlyn et al., Science DOI: 10.1126/science.aaf5037).

We thank the Referee for these questions as we acknowledge there were not addressed sufficiently clearly in our manuscript. Our derivation does not rely on having monopoles of unit charge, but it does require a node that is made only of two bands, so that Eq. (6) applies. SrSi₂ is one of these cases with $C = 2$, and quantization is expected. We have added two more references that discuss the existence of two-band nodes of higher monopole charge to emphasize this point, as well as an explanation of this fact in the text.

Regarding 1.3, quantization does not trivially apply to the novel fermions discussed by Bradlyn et. al., because the nodes are formed by more than two bands, and the corrections in Eq. 8 will always be present and cannot be made small by changing the frequency. Since this remains an interesting case for further study we have added a sentence to the main text and reference to this paper.

- 2. Experimental feasibility: As indicated from equation 1, the quantization still depends on the intensity of the laser, I . This is quite difficult to measure in experiment. It depends on multiple factors, the absorption coefficient, the thickness of the sample, the size of the beam spot, etc. Because the purpose is to measure a quantized value, it becomes important to determine I precisely. The authors should comment on this aspect.

This is indeed an aspect of the measurement that has to be carefully considered. Our prediction of a bulk quantized current is made assuming a constant intensity of light, equal to the external, applied intensity (which is known), but intensity will decay as light penetrates in the material due to absorption. However, this effect is actually negligible for Weyl semimetals, for which the attenuation length scales as $1/\omega$ at low frequency. We have now included an extra section in Methods where we compute the attenuation length from the conductivity of a Weyl semimetal, and show that for the frequencies of interest (up to 100 meV or 25 THz) the attenuation length is larger than $2\mu\text{m}$, so for the typical thin films used in experiments absorption is negligible. We have also included a statement in the main text explaining this. The intensity therefore remains constant through the sample and our

prediction for the quantized current density applies. As in any transport measurement there will be a geometrical factor relating current density to total current, but this is known for a given experiment and we believe should pose no problem.

- 3. Lastly, a minor issue. The proposal requires a Weyl semimetal with no mirror and no inversion. This type of Weyl semimetal has been proposed and systematically analyzed in Chang et al., <https://arxiv.org/abs/1611.07925> with more material candidates. The authors should consider citing this paper.

We thank the referee for bringing this very relevant reference to our attention. The current manuscript now mentions and cites this work (Ref.[46]) in the paragraph about candidate materials.

Referee B

- 1. The authors claim in the title and in the abstract that they have found a quantized circular photogalvanic effect in WSM. What they have actually found is a quantized value of a piece of the circular photogalvanic effect (CPGE): the photocurrent or injection current. There are other components that are part of the CPGE and they are taking a quantized value. While this statement is made clear throughout the body of the manuscript, it is not explicitly written in the title and abstract. It is mentioned that the trace part of the effect is quantized, but, for the benefit of the readers not being experts in these non-linear effect, it should be explicitly said in the title or abstract.

We thank the referee for this remark. We would first like to emphasize that in the monochromatic limit the shift contribution to the CPGE is zero, and only the injection part, characterized by the tensor β_{ij} defined in the text, remains finite. As the referee points out, our work only is concerned with the trace of this tensor, which is the one where quantization is found. We have now updated the abstract to explicitly reflect that it is only the trace of the injection part that shows this quantization. We agree that this should be so for the benefit of non-expert readers.

- 2. The authors compute the injection current within the dipole approximation, so there is momentum conservation together with energy conservation. It means that, when computing the coefficient β_{ii} with the Fermi golden's rule (or Floquet), the imaginary part comes from a resonant (or nesting) condition between the valence and conduction surfaces, and this precise sum over momenta is what allows the authors to find the integral over the Berry curvature (over the whole Fermi surface). They also mention a proposed material, SrSi₂. But looking at the bandstructure, it seems that the Weyl nodes are tilted, even very close to the Weyl node. This motivates this question: when tilting is unavoidable, in Weyl semimetals the optical conductivity at energies of two times the chemical potential does not take a quantized value (for isotropic WSM it does, leading to a quantized value of the conductivity at this energy). The reason for this is that, due to the tilting, the nesting condition is lost and the sum over momenta does not sweep the whole Fermi surface but just one dimensional lines, at best. This has been addressed recently (PHYSICAL REVIEW B 94, 165111 (2016)). My question then is, when tilting is unavoidable, if the same effect happens here, and the sum over the whole Fermi surface leading to the value of the topological charge disappears, leading to a non quantized value of β .

The referee raises an important point. Tilting will indeed modify the resonance condition, and this will change the shape of the resonant manifold in k-space where $\omega_{12}(k) = \omega$. In the language of the mentioned paper where w is the tilt parameter in units of v_F , this

leads to a frequency window $2\mu/(1+w) < \omega < 2\mu(1-w)$ where the resonant manifold is not a closed sphere, and this changes the value of the conductivity. However, the effect disappears for $\omega > 2\mu(1-w)$ where the resonant manifold becomes closed again. Regarding the CPGE integral then, the only effect of the tilt is to change the energy window over which quantization can be observed. However, as we stated in the paper, for type II nodes ($w > 1$ in this language) there is no way to find a closed resonant manifold at any energy and the quantization is generically lost. We have included a more explicit discussion of these facts and the mentioned reference (Ref.[51]) in the updated version of the manuscript.

- 3. Even for inversion symmetric systems, the presence of surfaces in real samples trivially breaks inversion symmetry, inducing second-order non linear optical responses. The question is if the presence of surfaces can contribute to the injection current leading to a non-quantized value of β . Can the effect of the surface be ruled out in the injection current part?

This is an interesting point that was not addressed in the previous version of the manuscript. There are two main reasons why surface states are not expected to modify our predictions. The most important one is that the quantized CPGE contribution comes from the diagonal part of the CPGE tensor, i.e. it comes from a current that is parallel to $\vec{E} \times \vec{E}^*$ and thus perpendicular to the polarization plane. At normal incidence, this current is perpendicular to the surface, and therefore cannot be carried by surface states. The other important reason is that the predicted CPGE current density is a bulk effect, generated throughout the bulk of the material (as argued before absorption and the decay of the intensity are negligible). The total current scales linearly with the thickness and any surface contribution would in any case be negligible compared to the bulk one.

- 4. A (really) minor stylist suggestion: When computing the injection current by using Floquet, the authors use the same notation for the velocity elements and the position of the Weyl nodes in the previous section. I suggest the authors to clarify the nomenclature.

We thank the Referee for pointing out this. In the revised manuscript, we adopted the same notation in the Floquet theory section for the energy dispersion and the velocity matrix element as in the previous sections. We clarified that we use the same notation for these quantities in the sentences above and below Eq.(26).

- 5. The lattice models the authors employ have been used and described much before the reference [49], for instance, in ref. [16] and many others. I would encourage the authors to be "fair" and make an effort to give credit to other publications discussing these models.

In the previous version of the text, we cited Ref. [16] in the main text and Ref. [49] (now [56]) in the Methods section. We now include Ref. [16] also in the Methods section to acknowledge its relationship with the lattice model used.

REVIEWERS' COMMENTS:

Reviewer #1 (Remarks to the Author):

The authors have very nicely addressed all questions I had. I recommend the revised manuscript for publication

Reviewer #2 (Remarks to the Author):

Dear editor,

After reading the questions sent by the authors to my questions and the answers given to the other referee, I suggest the present version for the manuscript adequate for being accepted in Nature Communications.

Concerning the question about the fate of quantization of β when tilting is considered, I'm satisfied with the answer. I was worried if the predicted phenomenon could be actually been observable in SrSi₂, and/or if the authors had used specific numbers for this particular material. Since they have provided other compounds as candidates (as requested by the other referee) there is no need to use realistic parameters of a very specific material, and now it is clear that the possibility of the discussed effect to be observed increases, with the identification of more material candidates.

Concerning the second question related to the presence of surfaces, I agree with the authors that the described effect comes from bulk, and surface states might not need to contribute as long as the sample is large enough (and bulk quantities are well defined).

Alberto Cortijo